# Identification and Preliminary Analysis of Granulosa Cell Biomarkers to Predict Oocyte In Vitro Maturation Outcome in the Southern White Rhinoceros (*Ceratotherium simum simum*)

**DOI:** 10.3390/ani14233538

**Published:** 2024-12-07

**Authors:** Elena Ruggeri, Kristin Klohonatz, Barbara Durrant, Marc-André Sirard

**Affiliations:** 1Reproductive Sciences, Conservation Science Wildlife Health, San Diego Zoo Wildlife Alliance, Escondido, CA 92027, USA; bdurrant@sdzwa.org; 2Center for Research on Reproduction and Women’s Health, Perelman School of Medicine, University of Pennsylvania, Philadelphia, PA 19104, USA; kristin.klohonatz@pennmedicine.upenn.edu; 3Département des Sciences Animales, Université Laval, Quebec City, QC G1V 0A6, Canada; marc-andre.sirard@fsaa.ulaval.ca

**Keywords:** wildlife, assisted reproductive technologies, oocytes

## Abstract

Granulosa cells play an essential role in oocyte meiotic competence acquisition by supplying nutrients and metabolites to oocytes and secreting paracrine signals that regulate oocyte maturation. Identifying biomarkers in granulosa cells to measure an oocyte’s ability to mature in vitro noninvasively predicts success or failure prior to maturation. This technique also offers the opportunity to develop treatment strategies to overcome potential maturation failures. This study aimed to evaluate potential biomarkers associated with follicle development, meiotic competence, cell death and atresia, and embryonic genome activation in granulosa cells from oocytes that did or did not mature after in vitro fertilization in the southern white rhinoceros. This study determined eight potential biomarkers associated with the success or failure of an oocyte to mature. Two genes were correlated with follicle development, three genes with meiotic competence, and three genes with cell death and atresia. This is the first study in which in vivo granulosa cells were used as a diagnostic predictor of oocyte maturation competence in the southern white rhinoceros. It represents a critical evaluation needed to improve assisted reproductive technologies in this species.

## 1. Introduction

Oocyte competence is the ability of the oocyte to complete both nuclear and cytoplasmic maturation, achieve fertilization, develop to the blastocyst stage, and terminate with a successful live birth [1]. Granulosa cells (GCs) play an essential role in oocyte meiotic competence acquisition by supplying nutrients and metabolites to oocytes and secreting paracrine signals that regulate oocyte maturation [2,3,4]. Understanding the mechanisms by which GCs influence oocyte developmental competence can help identify predictors of fertility as well as contribute to improving assisted reproductive technologies.

Traditional oocyte quality assessment has been based on morphology, but in recent years a multitude of biomarkers in granulosa cells and cumulus–oocyte complexes (COCs) have been determined and associated with oocyte maturation, in vitro fertilization success, and embryo development outcome [1,5,6]. The identification of noninvasive biomarkers of oocyte competence is one of the foremost targets of current research in reproduction and can aid in improving the development of in vitro matured oocytes [5]. In humans, biomarkers associated with competent oocytes have been identified and can be used for embryo selection [7,8,9].

FSH and LH stimulation regulate many of the GC biomarkers. Reduced response to these gonadotropins has been shown to downregulate genes associated with atresia and apoptosis, resulting in a compromised maturation process [10]. Therefore, different gene expression patterns in GCs may correlate with morphological and physiological characteristics, follicular environment, and fertility potential.

GCs are an easily accessible material that is often discarded during ovum pickup (OPU); hence, they are ideal samples with which to develop biomarker panels to establish oocyte competence potential. Transcriptomic, microarray, and gene expression analysis of GCs have been regarded as a tool for assessing oocyte quality and viability [11]. Furthermore, GCs have shown the ability to predict embryo development and pregnancy success [5,12]. Biomarkers of granulosa cells coupled with individual oocytes could be used in the assisted reproduction field to indicate which embryos have the best chance of implanting and concluding in a live birth. Although there is no one single gene transcript that appears to be capable of predicting which embryos produced by IVF will lead to live birth, numerous biomarkers in GC have been identified based on the endpoint to be evaluated, including oocyte nuclear maturation, fertilization, blastocyst morphology, implantation, and live birth. Although the most desirable endpoint for transcriptomic studies is live birth, several studies examined the gene expression profiles of GC influencing oocyte maturation and fertilization, as a proxy for live birth, as these are two mandatory steps that lead to embryo development and implantation [5,13]. By defining the expression profiles of specific genes in GCs associated with embryonic outcome we can develop treatment strategies to compensate for poor-quality oocytes deficient in specific genes during follicular development.

The southern white rhinoceros (SWR) is the closest relative to the functionally extinct northern white rhinoceros (NWR) [14]. Several groups worldwide have been working to develop assisted reproductive technologies for this species [15,16,17,18,19], mimicking well-established protocols and approaches effective in domestic species and humans. Our group has been largely focusing on studying SWR granulosa cells as a tool to study in vivo follicle dynamics and in vitro oocyte maturation requirements [18,19]. Because GC is an easily accessible material discarded after OPU, it represents a good biological sample for diagnostic research. By studying GC in SWR, it is possible to understand the molecular components driving oocyte competence acquisition in this species and aid in ARTs improvement. In addition, treatment strategies may be developed to compensate for poor oocyte quality due to the deficiency of specific proteins during follicular development.

This work aimed to identify biomarkers in GC from the SWR associated with oocyte in vitro maturation outcome. We evaluated GC collected from follicles that yielded oocytes that matured or did not after in vitro maturation culture. To reach our objective, we evaluated four major biological processes associated with oocyte maturation and developmental failure or success: follicle development, meiotic competence, cell death and atresia, and embryonic genome activation (EGA). For these analyses, we selected candidate genes from the literature in domestic species and humans as well as our previous work focused on GC transcriptome in the SWR [17]. For follicle development, the following eight genes were evaluated: collagen type I alpha 1 chain (*COL1A1*) [20], growth differentiation factor 9 (GDF9) [21], lysine acetyltransferase 8 (KAT8) [19], luteinizing hormone receptor (LHR) [22], mechanistic target of rapamycin (mTOR) [23], progesterone receptor (PGR) [24], tumor necrosis factor (TNF) [19], and tumor protein p53 (TP53) [25,26]. Cell death and atresia were assessed by measuring the following six genes: F-box and WD repeat domain containing 11 (*FBXW11*) [17], geranylgeranyl diphosphate synthase 1 (GGPS1) [27], junction mediating and regulatory protein p53 cofactor (*JMY*) [28], mevalonate kinase (MVK) [29], natriuretic peptide receptor 2 (NPR2) [30], and neuregulin 1 (NRG1) [30]. For meiotic competence, we evaluated the following three genes: collagen type IV alpha 1 chain (COL4A1) [31], macrophage immunometabolism regulator (MACIR) [17], and thymopoeitin (TMPO) [32]. For embryonic genome activation, we evaluated five genes: BCL2 related protein A1 (BCL2A1) [33], chaperonin containing TCP1 subunit 3 (CCT3) [34,35], heterogeneous nuclear ribonucleoprotein A2/B1 (HNRNPA2B1) [36,37], MYC proto-oncogene (MYC) [38,39,40], and nuclear transcription factor Y subunit alpha (NFYA) [18,41,42]. This study aimed to enhance our understanding of the supportive function of GCs in the SWR, leading to more targeted studies to improve oocyte maturation and embryonic development, with the overarching goal of advancing rhinoceros conservation efforts and ARTs improvements.

## 2. Materials and Methods

### 2.1. Animal Management and Ovum Pickup (OPU)

All procedures, experiments, and methods were reviewed and approved by San Diego Zoo Wildlife Alliance’s Institutional Animal Care and Use Committee (SDZWA IACUC; protocol number 21-016). The parous 12-year-old female southern white rhinoceros in this study was confirmed by regular ultrasound to be free of reproductive pathology. This female underwent ovarian stimulation before transrectal OPU as previously described [17,19,43].

Before OPU, the rhino received synthetic chlormadinone acetate (CMA) at 3 mg/day for 18 days (days 0–17). Forty-eight hours after CMA withdrawal (day 19), she received 1.8 mg of deslorelin [gonadotropin-releasing hormone (GnRH) analog] followed by 2.5 mg of deslorelin 48 h later (day 22; 4.3 mg total) via intramuscular injection. OPU was performed on Day 24. The female was anesthetized using a combination of etorphine (3.8 mg), medetomidine (49.6 mg), azaperone (30.5 mg), and butorphanol (49.6 mg) administered intramuscularly via remote drug delivery system. During initial positioning to facilitate intubation, propofol was administered intravenously (1000 mg). Anesthesia was antagonized with 248 mg atipamezole and 191 mg naltrexone administered intramuscularly resulting in a recovery without complications.

Following fecal removal, rinsing, and disinfection of the rectum, OPU was achieved using a customized, ultrasound-guided probe housing two double-lumen needles. Ovarian follicles were measured via ultrasound, then aspirated and flushed repeatedly with a warm (37 °C) flushing solution (Vigro) containing 12.5 I.U./mL of heparin. Follicular fluid and granulosa cells were separated based on follicle stage/size. Granulosa cells from dominant follicles (18–26 mm) were selected for this study.

### 2.2. Granulosa Cells Collection and RNA Isolation/Quantification

Next, OPU oocytes were retrieved from the collection fluid, evaluated and placed into in vitro culture. Free-floating mural granulosa cells (GC) were collected and pipetted directly into RNAlater (Thermo Fisher Scientific, Waltham, MA, USA) for 24 h at 4 °C. Multiple aliquots of GC from each follicle were then stored at −80 °C. For RNA isolation, the granulosa cells were thawed, mixed in equal parts with cold PBS (Sigma Aldrich, St. Louis, MO, USA) and centrifuged at 3000× *g* for 10 min. The supernatant was discarded, and the pellet was resuspended in cold PBS. The samples were centrifuged again at 3000× *g* for 5 min to remove all remnants of RNAlater and the supernatant was discarded again. Total RNA was isolated from granulosa cells using an Arcturus PicoPure RNA Isolation Kit (Thermo Fisher Scientific, Waltham, MA, USA) per the manufacturer’s instructions. Two follicles (biological replicates) were used for each group: polar body and no polar body outcome after in vitro maturation. Oocyte nuclear maturation completes primary meiotic division by extruding the first polar body, hence the polar body presence indicates metaphase II arrest and meiotic maturation completion. For each biological replicate, there were three individual technical replicates from individual GC tubes. Therefore, there was a total of *n* = 6 per group. Granulosa cells were incubated with extraction buffer for 30 min before centrifugation to remove debris and extracellular material. The cell extract was incubated with ethanol, then bound to, and washed on preconditioned purification columns. Total RNA was recovered into elution buffer and quantified using a Qubit 4 Fluorometer (Thermo Fischer Scientific, Waltham, MA, USA). After quantification, the samples were evaluated on a 4150 TapeStation System (Agilent Technologies, Santa Clara, CA, USA) to determine RNA quality.

### 2.3. cDNA Synthesis and Quantitative Real Time Polymerase Chain Reaction (qPCR)

Reverse transcription was performed using a QuantaBio qScript cDNA Synthesis Kit (VWR). Total RNA (3 ng/μL) was added to each reverse transcription reaction with 4 μL of reaction mix, 1 μL of reverse transcriptase, and nuclease-free water to reach a total reaction volume of 20 μL. cDNA synthesis was performed at 22 °C for 5 min, 42 °C for 30 min, 85 °C for 5 min and held at 4 °C.

Twenty-two genes were chosen for qPCR analysis based on the biological processes selected for this study. The genes selected based upon follicle development were collagen type I alpha 1 chain (*COL1A1*), growth differentiation factor 9 (GDF9), lysine acetyltransferase 8 (KAT8), luteinizing hormone receptor (LHR), mechanistic target of rapamycin (mTOR), progesterone receptor (PGR), tumor necrosis factor (TNF), and tumor protein p53 (TP53). The genes selected for meiotic competence were F-box and WD repeat domain containing 11 (*FBXW11*), geranylgeranyl diphosphate synthase 1 (GGPS1), junction mediating and regulatory protein p53 cofactor (*JMY*), mevalonate kinase (MVK), natriuretic peptide receptor 2 (NPR2), and neuregulin 1 (NRG1). The genes associated with cell death and atresia were collagen type IV alpha 1 chain (COL4A1), macrophage immunometabolism regulator (MACIR), and thymopoeitin (TMPO). The genes selected based upon their role in embryonic genome activation were: BCL2 related protein A1 (BCL2A1), Chaperonin containing TCP1 subunit 3 (CCT3), heterogeneous nuclear ribonucleoprotein A2/B1 (HNRNPA2B1), MYC proto-oncogene (MYC), and nuclear transcription factor Y subunit alpha (NFYA).

Our recent work on granulosa cells transcriptome analysis [17] provided a more complete and in-depth annotation of the northern white rhinoceros genome using SWR transcripts [44] that allowed the identification of genes not previously recognized in the southern white rhinoceros genome as biomarkers during oocyte development (COL1A1, JMY, MVK, FBXW11, NRG1, GGPS1, NPR2, TMPO, MACIR, COL4A1, BCL2A1, HNRNPA2B1, CCT3, MYC, and NFYA). Forward and reverse primers were designed using Primer3 version 4.1.0 (https://primer3.ut.ee, accessed on 11 March 2024) with a product size between 75–200 base pairs (bp), a primer length between 18–22 bp, a primer annealing temperature between 58–62 °C, and a GC content between 50–60%. Genes were selected based upon cell function, previous publications [19,43], or were identified as potential biomarkers for oocyte competence [17]. Genes, categories, and primer sequences are listed in Table 1.

Synthesized cDNA was utilized for quantitative real time PCR (qPCR). Bio-Rad iTaq Universal SYBR Green Supermix was used for each reaction. Each qPCR reaction contained 1.17 ng of cDNA, 2 μL of primer mix, 10 μL SYBR Green Supermix, and water to reach a total volume of 20 μL. Samples were loaded into 96 well MicroAmp Fast Optical Reaction Plates (ThermoFisher Scientific) and were run and analyzed in duplicate using QuantStudio6. Real Time PCR cycle conditions were per the manufacturer’s protocol: initial denaturation at 95 °C for 5 min, 40 cycles of denaturation at 95 °C for 15 s and annealing/extension at 60 °C for 60 s and melt curve analysis at 65–95 °C for 15 s at 0.05 °C per second. Cycle threshold (CT) values were normalized to an internal control, glutathione peroxidase 7 (GPX7).

### 2.4. Statistical Analyses

All statistical analyses were performed and graphed with GraphPad Prism (GraphPad Software version 10.4.0). For qPCR analyses, normalized CT values were utilized for statistical comparisons. An unpaired two-tailed student’s *t*-test was utilized using the Benjamini–Hochberg method to control for false discovery rates (FDR). For all statistical analyses, samples were considered statistically different at *p* ≤ 0.05. All data were transformed to 2^−∆CT^ for graphical representation (relative expression). Fold changes were calculated as log_2_ (PB/no PB). If the fold change was positive, it indicated that gene expression was higher in PB. If the fold change was negative, it indicated that gene expression was higher in no PB. PB represents GC from oocytes that matured after IVM by extruding the first polar body. No PB represents GC from oocytes that did not mature after IVM; therefore, no polar body was observed.

## 3. Results

Overall, 22 genes were evaluated for the following categories: follicle development, meiotic competence, cell death and atresia, and embryonic genome activation. Of these genes, nine were differentially expressed in in vivo granulosa cells collected before in vitro oocyte maturation (IVM), that ultimately culminated in polar body extrusion after IVM.

### 3.1. Granulosa Cell Biomarkers Linked to Follicle Development

Eight different genes (COL1A1, GDF9, KAT8, LHR, mTOR, PGR, TNP, and TP53) associated with follicle development were evaluated in in vivo granulosa cells. Of these eight genes, two were differentially expressed in granulosa cells from oocytes that did or did not mature by extruding a polar body after in vitro maturation (GDF9 and mTOR; *p* < 0.05). Both genes had higher expression values in samples from oocytes that matured (Figure 1).

### 3.2. Granulosa Cell Biomarkers That Lead to Meiotic Competence

Of the eight genes selected for this category, two (GGPS1 and JMY) demonstrated significantly higher expression in in vivo granulosa cells from oocytes that matured after in vitro maturation culture and one (NPR2) showed significantly more expression in in vivo granulosa cells from oocytes that did not mature (Figure 2).

### 3.3. Granulosa Cell Biomarkers Prophesying Cell Death and Atresia

Three different genes (COL4A1, MACIR, and TMPO) associated with cell death and atresia were evaluated in in vivo granulosa cells. All three of these genes were differentially expressed in granulosa cells from oocytes that did not mature after in vitro maturation culture (Figure 3).

### 3.4. Granulosa Cell Biomarkers Predicting Embryonic Genome Activation

Figure 4 contains the genes and fold changes associated with embryonic genome activation. Of the five genes selected for this category, NFYA was the only one significantly more expressed in in vivo granulosa cells from oocytes that did not mature after in vitro maturation culture.

## 4. Discussion

The goal of this study was to determine if granulosa cells (GC) collected in vivo can predict oocyte maturation in vitro in the southern white rhinoceros (SWR). Evaluating GC offers a non-invasive method to study the bidirectional communication between GC and the oocyte, a very important aspect for oocyte competence acquisition and prediction of embryo developmental potential. This novel approach has not been evaluated in this species and it could provide a beneficial tool to (1) determine if the pre-OPU ovarian stimulation protocol supports oocyte meiotic resumption and (2) supplement maturation media formulation and realistically predict maturation outcomes.

First, we evaluated genes associated with follicle development obtained from our previous studies [17,18,19]. Although eight genes were considered for this category, mTOR and GDF9 were the only two remarkably altered in granulosa cells obtained from oocytes that matured in vitro. Mammalian target of rapamycin (mTOR)-dependent pathways are prerequisites for processes that promote the completion of meiosis and are essential for the maintenance of oocyte genomic integrity, sustaining ovarian follicular development, and embryo development [23]. The expression of mTOR is associated with signaling components in GC and folliculogenesis [45]. mTOR signaling regulates GC proliferation in response to FSH stimulation [46] and directly regulates meiotic processes [23,47]. Growth differentiation factor 9 (GDF9) was more highly expressed in granulosa cells from oocytes matured in vitro compared to those that did not mature. GDF9 is an important oocyte-derived factor that regulates ovarian function in female reproduction, modulating both the fate of granulosa cells and the developmental competence of the egg. High levels of GDF9 are associated with oocyte maturation and embryo quality [21]. In our data, mTOR and GDF9 expression in GC obtained from follicles associated with oocytes that matured in vitro showed a positive correlation. We can therefore conclude that these two genes are biomarkers for the potential successful in vitro maturation of rhinoceros oocytes.

Through granulosa cell sequencing [17], we identified potential biomarkers associated with meiotic competence. Of the six biomarkers evaluated, three were significantly differentially expressed. GGPS1 and JMY exhibited significantly greater expression in the GC from oocytes that matured in vitro, and NPR2 was more highly expressed in GC from oocytes that did not mature. GGPS1, geranylgeranyl diphosphate synthase 1, is a crucial enzyme in the mevalonate pathway, which is responsible for cholesterol biosynthesis, cell growth and differentiation, and protein synthesis [27]. Cholesterol has an essential role in mitochondrial function, which is critical during oocyte maturation [48,49]. Steroidogenesis is driven by cholesterol biosynthesis and LH elevation, as seen in late folliculogenesis, which then results in decreased levels of cholesterol, which activates genes associated with cholesterol biosynthesis (i.e., GGPS1) [48,49]. The literature shows that females with decreased GGPS1 exhibit poor quality oocytes and meiotic abnormalities due to impaired mitochondrial function [27,48]. These data support our findings that GGPS1 levels were decreased in GC associated with oocytes that did not mature in vitro, compared to those that matured. This indicates that GGPS1 could be an effective early marker present in in vivo GC associated with the oocyte’s ability to mature in vitro.

Junction mediation and regulatory protein p53 cofactor (JMY) was decreased in oocytes that did not mature. This gene is highly involved in actin nucleation during oocyte polarization and affects the microtubule and microfilament cytoskeleton by activating Arp 2/3 [50,51,52]. It has been observed that if JMY is not expressed the spindle fails to migrate to the cortex, and the oocyte arrests with a centrally located spindle [53]. Therefore, JMY is required to be high during the early stages of oocyte maturation and folliculogenesis, but decreases before ovulation, as observed in our previous study [17]. The oocytes and granulosa cells for this study were collected from dominant follicles, which is the stage before the decrease in JMY (in pre-ovulatory follicles). JMY is a possible candidate to determine an oocyte’s developmental potential, as it changes its expression during follicle development. Finally, natriuretic peptide receptor 2 (NPR2) works synergistically with natriuretic peptide precursor type C (NPPC) to ensure proper cumulus expansion during oocyte development and guarantee meiotic arrest [54]. During folliculogenesis, FSH levels increase as the follicle continues to develop. Increasing FSH levels result in decreasing NPR2 levels [55,56]. Final NPR2 downregulation occurs due to the LH/amphiregulin/EGFR signaling pathway, which is highly active in late antral (dominant) follicles [57]. In addition, in culture, amphiregulin also downregulates the NPPC/NRP2 pathway and could be used to initiate meiotic resumption and oocyte maturation in vitro [57]. In our data, NPR2 was higher in GC associated with oocytes that did not mature in vitro, suggesting that these oocytes were still inhibited from resuming meiosis. GGPS1, JMY, and NPR2 could be used as gene candidates to assess meiotic competence, a complex yet fundamental process required to successfully develop assisted reproductive technologies (ARTs).

Genes involved in cell death and atresia were also evaluated in our study. The following genes were chosen from previous studies performed in our laboratory [17,18,19]. All three genes evaluated (COL4A1, MACIR, TMPO) were consistently highly expressed in granulosa cells from oocytes that did not mature; hence, their expression was directly associated with oocyte maturation failure. Collagen type IV alpha 1 chain (COL4A1) is a collagen gene found in the basement membrane of ovarian follicles [58]. In healthy growing follicles, COL4A1 expression decreases as folliculogenesis progresses, resulting in the proliferation of GC and follicle remodeling [59]. In addition, high levels of COL4A1 are exclusive to atretic follicles [2]. In humans, oocytes that undergo recurrent oocyte maturation failure (ROMA) have low levels of PIWI-interacting RNAs (piRNAs), that down regulate genes involved in the oocyte’s extracellular matrix [60,61]. One gene in particular affected by the piRNAs is COL4A1 [60]. Decreased piRNAs result in increased COL4A1 and ultimately the developmental arrest of the oocyte during maturation [60]. Another gene of interest associated with negative oocyte maturation outcome was macrophage immunometabolism regulator (MACIR). Previous studies have shown MACIR was highly expressed in oocytes that did not mature after in vitro culture [17]. MACIR (formally known as C5ORF30) is associated with tissue damage and modulates the immune response, regulating macrophage function [62,63,64]. In addition, studies have shown that in cattle granulosa cells, MACIR is upregulated in follicles undergoing late-stage atresia [26]. Finally, thymopoeitin (TMPO), also known as lamina-associated polypeptide 2 (LAP2) [65], was also upregulated in granulosa cells from oocytes that did not mature in vitro. According to previous studies, TMPO should decrease throughout follicle growth progression [2,32], resulting in low levels of LIN28A [66,67]. LIN28A levels should remain low until after meiosis resumption, and in mice, high LIN28A levels have been directly associated with MII-arrested oocytes [66,68]. The combination of these three genes, COL4A1, MACIR, and TMPO being more highly expressed in granulosa cells from oocytes that did not mature in vitro is an indicator that prior to OPU these oocytes were already primed for failure to mature. In conclusion, our findings suggest that COL4A1, MACIR, and TMPO could be excellent candidates to determine if the follicles were undergoing atresia at the time of ovum pickup (OPU) and if the oocytes collected had already undergone irreversible atretic commitment. In both horses and cattle, follicle growth begins with increasing FSH levels due to pulsatile GnRH stimulation [69,70]. This results in multiple follicles growing but only one reaches the dominant size (18–26 mm in horses [71] and 10 mm in cows [72]). The subordinate follicles then plateau when a dominant follicle is selected and eventually all the nondominant follicles begin to undergo atresia [70]. For this study, all oocytes and granulosa cells were collected from dominant-size follicles, but some oocytes matured while others did not. This indicates that the oocytes could have been collected after the plateau phase and some follicles had already started to become atretic. Other studies [73], report performing OPU 24 h after the last GnRH stimulation, whereas in this study, OPU was performed 48 h after the last GnRH stimulation. We hypothesize that the timing of oocyte collection could be one of the factors contributing to the final oocyte maturation outcome. Modification of the ovarian stimulation protocol and analyzing cells at different follicle stages are areas of interest to be further addressed.

For the two oocytes that matured in vitro, intracytoplasmic sperm injection (ICSI) was performed successfully, but both embryos arrested at the four-cell stage. Embryonic genome activation (EGA) is a crucial event in embryo development when the transcriptional programming of the embryo is initiated and the developmental control switches from maternal to zygotic, but the timing is species specific [74,75]. In horses and humans, EGA occurs between four to six cells [76,77], but in cows it happens between eight to sixteen cells [78]. It is a mystery when this major genomic event occurs in the SWR, and there is no literature on this matter. As the domestic horse and SWR are taxonomically close [69], we can hypothesize that perhaps EGA may occur around the four to six cell stage. This would support the idea that the two fertilized oocytes could have arrested during their early embryo development due to EGA failure. Previous work in humans focused on identifying genes expressed in GC that could be used as biomarkers for oocyte and embryo quality linked to EGA [6,12,13,79]. Hence, five genes associated with EGA in horses, humans, and cows [35,36,40,41,42,75,80] were evaluated in this study. One gene, nuclear transcription factor Y subunit alpha (NFYA), was more abundant in GC from oocytes that did not mature. NFYA is the regulatory subunit of the NFY complex, and when it binds to DNA it forms a histone-like structure promoting chromatin accessibility [41,42]. In addition, NFYA is essential for activating a subset of genes during the major wave of EGA and it is required for embryos to develop to the blastocyst stage [40]. NFYA should be highly expressed in embryos that underwent EGA, and low levels should be observed in arrested early embryos. Our data showed very low levels of NFYA in GC associated with oocytes that matured but arrested at the four-cell stage. Unexpectedly, we observed very high expression levels of NFYA in granulosa cells associated with oocytes that did not mature after IVM. Although only NFYA was significantly different in our data, it is important to further evaluate EGA timing and targeting more genes involved in this crucial developmental point. Gene expression analysis related to EGA in in vivo GC could provide a non-invasive assessment for identifying the most competent oocytes capable of successful fertilization, passing EGA, and completing embryonic development. We hypothesize that perhaps due to the species specificity of this event, the most critical genes for EGA may be different in the SWR compared to domestic species and humans.

It is important to note one limitation of this study, which is that all the follicles were obtained from the same female and one OPU procedure. We attempted to overcome this by using multiple biological replicates (follicles) with multiple technical replicates within each specific follicle. As OPU in this species is not often performed, our sample size was limited, and we hope to generate a greater library of samples to continue evaluating this biomarker analysis.

In summary, GC collected in vivo can be a valuable non-invasive approach for evaluating the follicle processes associated with oocyte quality and its developmental potential. By screening in vivo GC for the identified biomarkers, realistic allocation of laboratory resources can be applied on an individual oocyte basis. In a species with limited opportunities to collect oocytes, it is important to have a realistic approach and expectation for scientific investigation. If biomarker analysis predicts the developmental potential of an oocyte, immediate adjustments to the in vitro maturation system can be applied or nonviable oocytes can be allocated to investigative research. Continuous analysis of these easily accessible (and often discarded) granulosa cells will improve the understanding of multiple aspects of reproductive biology in this species, which will in turn have an impact on conservation.

## 5. Conclusions

In conclusion, this study provides novel information on gene expression associated with oocyte in vitro maturation outcome in the southern white rhinoceros (SWR). This study offers potential biomarkers for clinical oocyte quality assessment and information to develop better ovarian stimulation protocols and improve in vitro maturation media formulation. To our knowledge, this is the first time in vivo granulosa cells have been used as a diagnostic predictor of oocyte maturation competence in the SWR, a critical evaluation needed to improve assisted reproductive technologies in this species.

## Figures and Tables

**Figure 1 animals-14-03538-f001:**
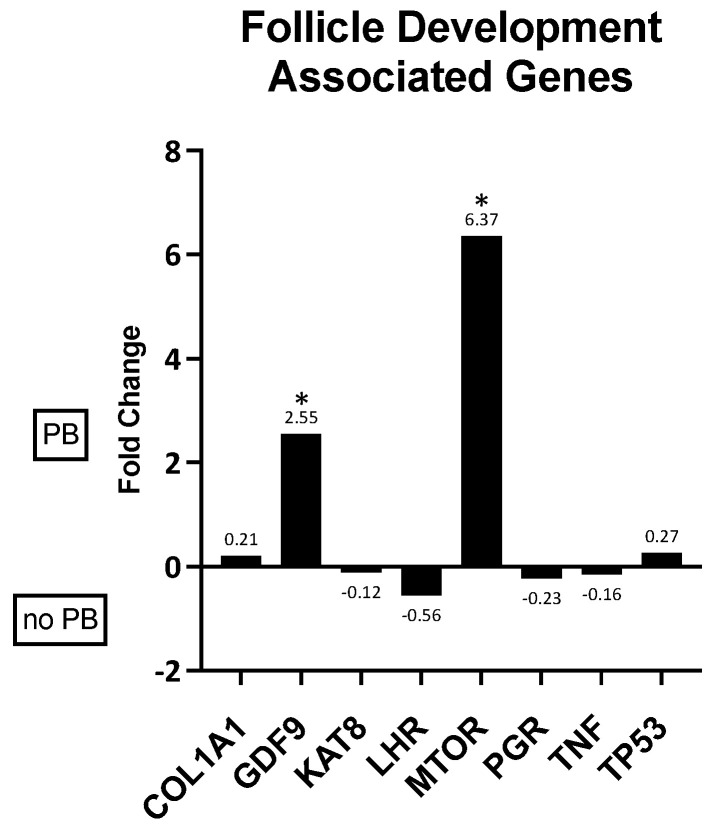
Fold changes in genes associated with follicle development in granulosa cells associated with oocytes that did or did not mature in vitro. A positive fold change indicates gene expression was higher in cells from oocytes that matured (PB) while a negative fold change indicates gene expression was higher in cells associated with oocytes that did not mature (no PB) after in vitro culture. * *p* ≤ 0.05.

**Figure 2 animals-14-03538-f002:**
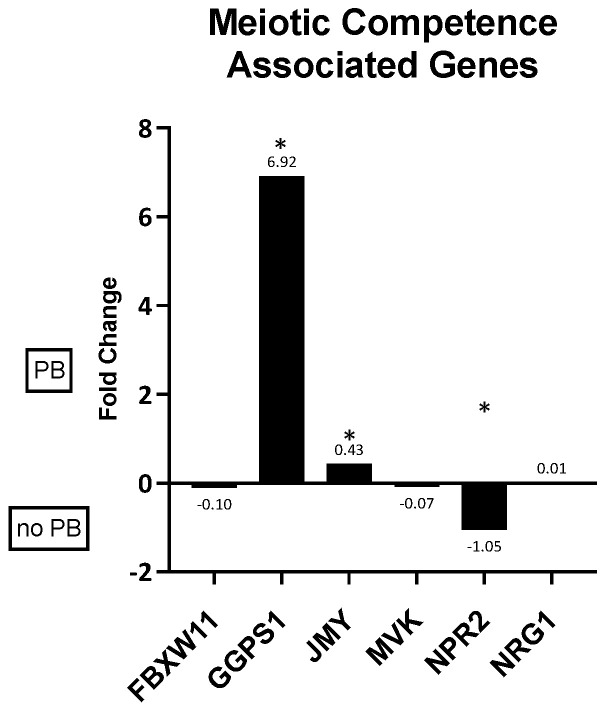
Fold changes in genes associated with meiotic competence in granulosa cells associated with oocytes that did or did mature in vitro. A positive fold change indicates gene expression was higher in cells from oocytes that matured (PB) while a negative fold change indicates gene expression was higher in cells associated with oocytes that did not mature (no PB) after in vitro culture. * *p* ≤ 0.05.

**Figure 3 animals-14-03538-f003:**
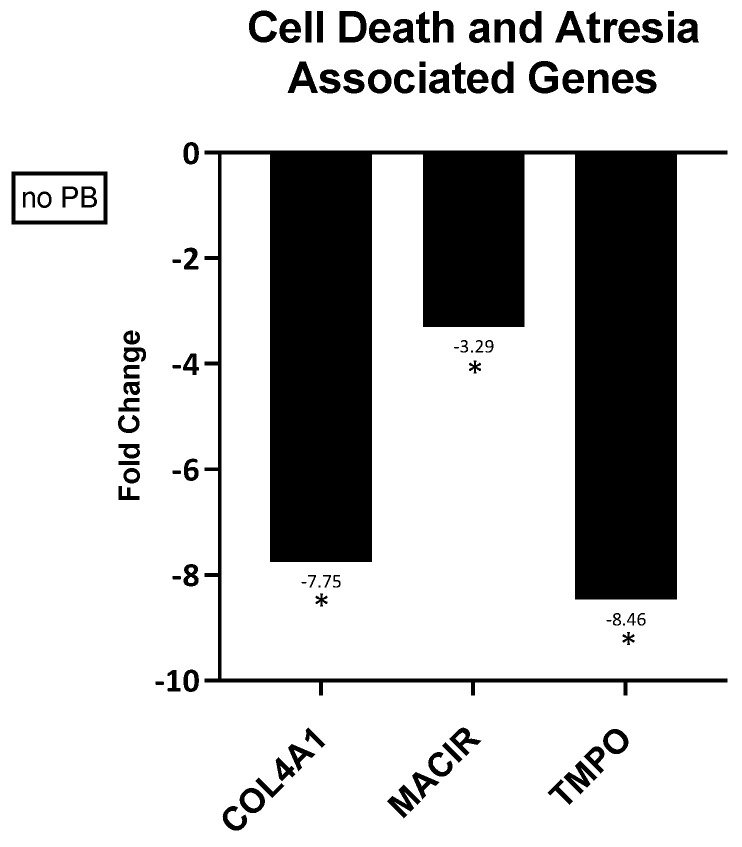
Fold changes in genes associated with cell death and atresia in granulosa cells associated with oocytes that did or did mature in vitro. A negative fold change indicates gene expression was higher in cells associated with oocytes that did not mature (no PB) after in vitro maturation culture. * *p* ≤ 0.05.

**Figure 4 animals-14-03538-f004:**
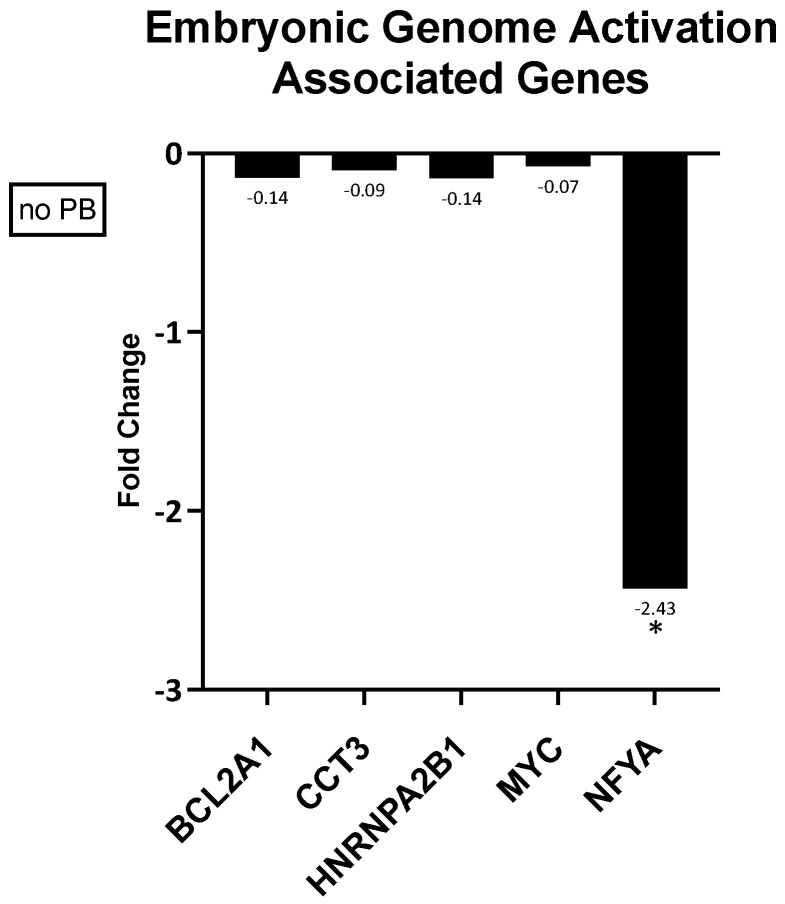
Fold changes in genes associated with embryonic genome activation in granulosa cells associated with oocytes that did or did mature in vitro. A negative fold change indicates gene expression was higher in cells associated with oocytes that did not mature (no PB) after in vitro maturation. * *p* ≤ 0.05.

**Table 1 animals-14-03538-t001:** This table contains the gene name, category, and forward and reverse primers used for quantitative real-time PCR.

Gene	Category	Forward Primer (5′-3′)	Reverse Primer (5′-3′)
COL1A1	Follicle development	GCATGGCCTGATTAGCAGTG	GCAGTTAGGTTCGCGTGTTC
GDF9	Follicle development	ACCAGGTGACAGGAACCGT	CAGCTCTAGGGAGAGTCTTGC
KAT8	Follicle development	CCTCCTGACACGTCACAGAC	ATCGCTGTGGTGGAAGTGAC
LHR	Follicle development	TGGAAGTGATAGAGGCGAACG	GTTCTGGAAGGCATCAGGGT
mTOR	Follicle development	GGCAGCATTAGAGACAGTGGA	AATCGGGTGAATGATCCGGG
PGR	Follicle development	TCCCACGAACGTAGAGAGGC	TGAACAGTCCCCAATGTGGC
TNF	Follicle development	GCCCATGTTGTAGCAAACCCT	AGGAGCACATGGGTGGAAGA
TP53	Follicle development	AGGTACGTGTTTGTGCCTGT	TCACGCCCACGAATCTGAAG
FBXW11	Meiotic competence	GCACACAGAGACCTGGCATC	GGTACGTTTCCACGTTGCCT
GGPS1	Meiotic competence	ATACCGCTTGTCAGGCCATC	ATCCATGCCAATCCCCCTCT
JMY	Meiotic competence	GAACTTGCCATGCTACGACG	CTTTGGGGAGAAAGGAGCAGA
MVK	Meiotic competence	TATACCCCGAGTTCCTGGCA	CTTGGCTGCTCAGTCCGTTA
NPR2	Meiotic competence	GGCACCCTGAGAAAGCATCC	GGGTGGTTGATAGGTTAGGGC
NRG1	Meiotic competence	TTACTTCGTGGAACCCGTGG	GAGGGGCCTTTCAGATGACC
COL4A1	Cell death and atresia	TCCGTTTGCTTGCTTTGCTC	TCAGGGTTTGAAGCTCCGTC
MACIR	Cell death and atresia	AAATCACAGGTACTCGGGGG	GTCAGCAACAACCAAGCAGA
TMPO	Cell death and atresia	TCGCCCTAAGCCTAACATCTG	GTGGCAGTGGCTCACATAGA
BCL2A1	Embryonic genome activation	CCAAATCTGGCTGGCTGACT	GGCAGTTTTCCCAAGATGGA
CCT3	Embryonic genome activation	TCTTTGCTGGACCCCTGAAG	AGGAGCAAGCCTGTTGGAAA
HNRNPA2B1	Embryonic genome activation	GGCTAAGGAAAAGGTAGGGGC	ATGGTAGGGGATTGGGGAAGA
MYC	Embryonic genome activation	TCCTCTTCTTATTGGCGGCT	TCTAAGGGGAAGGGATGGGA
NFYA	Embryonic genome activation	ATACCTGCATGAGTCTCGGC	GGTACAAGTCTTCTCACCTGC

## Data Availability

Data is contained within the article or Appendix A.

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
