# Peer review of "Identification and Preliminary Analysis of Granulosa Cell Biomarkers to Predict Oocyte In Vitro Maturation Outcome in the Southern White Rhinoceros (Ceratotherium simum simum)"

_animals, 2024, doi:10.3390/ani14233538_

Round 1

Reviewer 1 Report

Comments and Suggestions for Authors

I very much liked the manuscript and the approach. While the very impressive results shown in Fig. 3 were rather expected, the likewise very impressive results shown in Fig 4. (erroneously also called Fig. 3) were not expected. At least not by me. May this also have something to do with the 4-cell stage arrest?

I have some formal comments and some questions.

L58:  please rephrase „reproductive field“ (the authors are likely not talking about a field that is reproductive)

L69: Even though I have seen it in other papers too, I nevertheless find the expression “noninvasive tool” somewhat misleading if GCs are collected during ovum pickup. Although technically correct (as no epidermis is being penetrated), the procedure as such is not as stressless for the animal like shedding hairs or faeces. So, my suggestion is to remove “noninvasive” (it is not needed for the statement of the paper anyway). In addition, I also suggest to remove the word “potential”, because GC have been regarded as tools and not as potential tools.

L70. Here, two things are being mixed. Up to this sentence, arguments were directed at oocyte maturation. Here, all of a sudden, the authors introduce the “reproductive potential of embryos”, which is something completely different from oocyte maturation. This needs to be clarified (e.g. by a transitory sentence). This may also help the reader to better understand the different assessment endpoints mentioned on L76. As embryonic genome activation (LO98) will be one of the four biological processes assessed in this manuscript, the reader should be guided to that.

L71-81. Here, an explanatory sentence may help pointing out that oocyte maturation will thus be an endpoint proxy for live birth

L86: my suggestion to remove “noninvasive” remains (here it is written non-invasive)

L89. “the” in front of SWR may be removed

L89. Not sure that “understanding the molecular mechanisms” is the correct expression. Isn’t it rather identifying some of the components of the mechanisms?

L92: “deficiency of specific genes” sounds weird to me. What is meant by that? Expression deficiency? And what is then meant with “treatment”? Please explain.

L100: please rephrase “sequencing of GC” (perhaps sequencing of CG genomes/transcriptomes/ …?)

L100: please add the number “eight” in front of “genes” (the mentioning of the number of genes assessed for the different biological processes becomes important on L160, where all of a sudden “twenty-two genes” are mentioned [8+6+3+5])

L101-102: as not every interested reader will be familiar with this gene nomenclature, please provide the full name first and then the abbreviation code. The authors have done this on page 4 (L162-174), but I think it should be done here, as the genes are mentioned here for the first time. To facilitate downstream reading I would leave the explanations an page 4 too.

L102: please add “the following six genes” after “by measuring”

L104: please add “three genes” after “the following”

L105: please rephrase in active voice: “we evaluated the five genes…” (removing “were evaluated” at the end)

L106: I find the word “reproductive” misleading here. Perhaps better: “supportive”.

L108-109: just a suggestion: not every reader might be aware of the high costs (money, resources, manpower) associated with ART, which is all wasted with a non-live birth outcome. Thus, increasing the chances for a live birth outcome will reduce any such waste. So, perhaps pointing this out in an additional sentence may help the readership to understand better.

L127: I suggest to replace “uncomplicated” by “without complications”

L128: please add: “rinsing and disinfection” of what?

L139: I know what is meant, but it is technically incorrect. Technically correct would be 1:2. If the authors want to avoid confusion for people not familiar with the concept, than I suggest to use “mixed in equal parts with PBS”

L142: please add: “and the supernatant was discarded again” in front of “to remove …”

L145: I suggest to add a sentence or half sentence as to why “formation of a polar body” was chosen as end point.

L153: somewhat confusing. As quantification was already carried out on the Qubit, the use of the TapeStation for another quantification seems unnecessary. My guess is that is was used to determine the molecular size range (to assure large enough cDNAs for qPCR)

L156: please provide RNA concentration or concentration range

L161: my suggestion is to stick to the four biological processes mentioned before (L97-98), to better allow the reader to make a connection to what was said before

L165: Again a suggestion. I would keep the order and the names of the processes always the same throughout the paper to avoid reader confusion (on page 2 the order is: follicle development, meiotic competence, cell death and atresia, and embryonic genome activation (L97-98). On page 4 (L161-174) the order is: follicle development, meiotic competence, cell death and atresia, early embryonic development [here even the name of the process changes].

L175-176: what does “These” refer to? If the authors mean that 8 out of 22 primer pairs had already been used in a previous study, they should say so (although I do not see the importance for this manuscript that 8 out of the 22 primer pairs had already been used in different study)

L176: please correct phrasing (sequencing GC transcriptomes or whatever was sequenced, but not “sequencing cells”)

L176-177. Odd phrasing. Not clear, if SWR transcriptome date were used to improve the annotation of the Northern White Rhino genome. Please specify, if SWR GC transcriptomes were used for NWR genome annotation improvement or if NWR GC transcriptomes were used.

L206-208. I may have missed it, but so far the grouping was “formation of polar body” and “no polar body” (L145). The reader does not automatically know what is group A and what is group B. My guess is that group A are oocytes that matured and had a polar body while group B are the ones that did neither.

L220: please remove last sentence and put “Fig. 1” in parentheses after “matured”.

Fig. 1: please explain the abbreviations at the Y-axis (PB and no PB). Although my suggestion is to use Group A and Group B instead, because this could then also be set in brackets in the legend text after the corresponding explanations [… was higher in oocytes that matured and extruded a polar body (group A) ….]

L228: Please omit this sentence. The reference to Fig.2 should be at the end of the paragraph in parentheses.

Fig. 2: same as in Fig 1.

Fig. 2: As we are talking log2-fold changes on N=6, I find 0.43 and -1.05 not very strong for a general conclusion

L243: again, please remove the last sentence and put “Fig. 3” in brackets after “culture”.

L258: I guess this should be Fig. 4.

Discussion should be shortened (3 full pages!)

The discussion is essentially divided in two parts. One part discusses the actual results from the aims of the study using the end point “formation of a polar body”.

The second one (quite unexpected by the reader) deals with the “continuation of the process” (starting L393). Here the authors discuss the continued fate of matured oocytes that were subsequently fertilized. In the case of the two oocytes concerned, they did not continue to develop into a blastocyst but went into cell division arrest. This information is certainly of valuable to the reader, but it should clearly be put under a separate header as it is only loosely connected to the rest of the manuscript.

I agree with the authors that the fact that all follicles were obtained in the same procedure from the same female rhino, which may limit (at the moment) the generalizability of the results.

What I am missing, is some sort of guidance, how these results can be used in the future. My point being that here two groups have been compared and x-fold differentiation in several genes has been measured (knowing the outcome: maturation with polar body or no maturation). The final goal would be to have some sort of threshold in gene expression that can be used by other researchers to assess the likelihood that the oocyte planned to be used is a good one.

Reviewer 2 Report

Comments and Suggestions for Authors

Ruggeri et al. present a novel analysis of gene expression in granulosa cells from the Southern white rhinoceros.  As would be expected from this species, research material was limited, which limited the conclusions that can be drawn.  However, this is the only data available for this species, so even limited data has great value.

The authors thoroughly describe all methods and do a good job of presenting both the limitations and novelties of their results.

My only request would be a more clear description of the number of OPU procedures performed and the number of oocytes recovered in each.  It sound like they had 4 oocytes, of which 2 matured and 2 did not.  Is that correct?  
